# The Impact of Implementation of Palliative, Non-Operative Management on Mortality of Operatively Treated Geriatric Hip Fracture Patients: A Retrospective Cohort Study

**DOI:** 10.3390/jcm13072012

**Published:** 2024-03-29

**Authors:** Thomas Nijdam, Tim Schiepers, Duco Laane, Henk Jan Schuijt, Detlef van der Velde, Diederik Smeeing

**Affiliations:** Department of Trauma Surgery, St. Antonius Hospital Utrecht, 3543 AZ Utrecht, The Netherlands

**Keywords:** hip fracture, geriatric, mortality, complications, palliative, non-operative, operative

## Abstract

(1) **Background:** Hip fracture patients with very limited life expectancy can opt for non-operative management (NOM) within a palliative care context. The implementation of NOM in the palliative context may affect the mortality of the operatively treated population. This retrospective cohort study aimed to determine whether the operatively treated geriatric hip fracture population would have a lower in-hospital mortality rate and fewer postoperative complications after the introduction of NOM within a palliative care context for patients with very limited life expectancy. (2) **Methods:** Data from 1 February 2019 to 1 February 2022 of patients aged 70 years or older were analyzed to give a comparison between patients before and after implementation of NOM within a palliative care context. (3) **Results:** Comparison between 550 patients before and 485 patients after implementation showed no significant difference in in-hospital or 1-year mortality rates (2.9% vs. 1.4%, *p* = 0.139; 22.4% vs. 20.2%, *p* = 0.404, respectively). Notably, post-implementation, fewer patients had prior dementia diagnoses (15% vs. 21%, *p* = 0.010), and intensive care unit admissions decreased (3.5% vs. 1.2%, *p* = 0.025). (4) **Conclusions:** The implementation of NOM within a palliative care context did not significantly reduce mortality or complications. However, NOM within palliative care is deemed a more patient-centered approach for geriatric hip fracture patients with very limited life expectancy.

## 1. Introduction

Hip fractures in older patients are increasingly prevalent in current trauma care and create a rising problem, with the global number of hip fractures expected to increase to 6.3 million annually in 2050 [1,2]. Hip fractures are historically treated with operative management (OM); however, outcomes (i.e., mortality and morbidity) following OM remain very poor for specific phenotypes of geriatric patients with several risk factors associated with adverse outcomes [3,4,5,6]. These risk factors include increasing age, the American Society of Anesthesiologists (ASA) classification of three or higher, low ambulation status, and cognitive impairment [4,6]. For these frail geriatric hip fracture patients with very limited life expectancy, a new non-operative management (NOM) within a palliative care context is increasingly offered as an alternative to OM [7,8]. This differs from historical non-operative management, which was typically received by relatively young and healthy individuals with non-displaced femoral neck fractures, or those with significant comorbidities that rendered surgery non-beneficial, often leading to surgical refusal by the anesthesia team [7,9]. Therefore, NOM within a palliative care context encompasses a distinct form of non-operative treatment, different from the historical non-operative approach. With NOM within a palliative care context, patients can opt through shared decision-making (SDM) for a more peaceful last phase of life compared to an uncertain period of invasive rehabilitation after hip fracture surgery [10,11]. Current literature has focused on the mortality of the geriatric hip fracture population, including operatively treated patients with very limited life expectancy [12,13,14,15]. However, these mortality rates for geriatric hip fracture patients after OM could be skewed due to this group of frail patients with limited life expectancy. The hypothesis is that these patients, who only recently have other options than OM, influenced the previously reported mortality rate of the operatively treated hip fracture population and consequently made it appear worse. Therefore, after the introduction of NOM within a palliative care context, the operatively treated hip fracture population might show fewer adverse outcomes after OM, resulting in a decrease in the burden of care at the trauma-geriatric ward. This study aims to determine whether the operatively treated geriatric hip fracture population will have a lower in-hospital mortality rate and fewer postoperative complications after the introduction of NOM within a palliative care context.

## 2. Materials and Methods

### 2.1. Study Design

This retrospective cohort study was performed on geriatric hip fracture patients who presented to the emergency department (ED) of a large regional teaching hospital in the Netherlands between 1 February 2019 and 1 February 2022. Patients were identified from the electronic medical records through Diagnosis Related Groups (abbreviated as Diagnose Behandel Combinatie in Dutch): 218, hip fracture. The Business Intelligence department utilized code 218 to extract patient identification codes for all hip fracture patients. Patients were eligible for inclusion if they were aged 70 years or older and were admitted to the trauma-geriatric ward after OM for a hip fracture. Patients with a pathological hip fracture, an injury severity score of 16 or higher, or a periprosthetic hip fracture were excluded. File searching was conducted on all eligible patients to gather variables essential for this study. The STROBE guidelines were used to guide this study [16].

### 2.2. Non-Operative Management within a Palliative Care Context

NOM within a palliative care context has emerged as a novel treatment approach with a distinct palliative perspective. Loggers et al. have demonstrated that patients opting for NOM within a palliative care context experience a quality of life that is non-inferior to that of individuals opting for surgical intervention during the terminal phases of life; indicating NOM as a viable treatment strategy within palliative care settings, emphasizing a more patient-centered approach [11].

NOM within a palliative care context was introduced in this center as an option on 1 August 2020 for geriatric hip fracture patients considered frail and with very limited life expectancy. Patients were considered frail with one or more frailty criteria (body mass index (BMI) of 18.5 kg/m^2^ or lower, functional ambulation category (FAC) of 2 or lower pre-trauma, ASA score of 4 or 5) or on the clinical judgment of the attending surgeon when thought of limited life expectancy without meeting the frailty criteria [11]. This clinical judgment is informed by the intuition of highly experienced trauma surgeons who frequently encounter such patients. The option is presented through an extensive shared decision-making (SDM) conversation, frequently involving family members, wherein both the outcomes of surgery and NOM within a palliative care context are thoroughly discussed. Approximately 25% of patients presenting to the ED engage in an extensive shared decision-making conversation, during which both treatment options are thoroughly discussed. Ultimately, approximately half of these patients choose NOM within a palliative care context, while the remaining half express a preference for surgical intervention. Hence, patients always retain the option to undergo surgery if they wish to. With NOM within a palliative care context, specific attention is paid to analgesia and patient comfort without aiming the patient to regain mobility and start the active rehabilitation program [11]. With the emergence of new effective analgesic modalities, such as the pericapsular nerve group block, a peripheral nerve block, satisfactory results regarding patient comfort ought to be achieved [17]. Since NOM within a palliative care context is not curative management, patients are likely to die within weeks after hip fracture (median survival 11 days (IQR 4-26)) [10]. The renewed hip fracture pathway for geriatric patients is shown in Appendix A.

### 2.3. Study Variables

The following baseline characteristics were collected from electronic medical records: age, sex, prior diagnosis of dementia (diagnosed by a geriatrician or general practitioner), BMI, Charlson Comorbidity Index (CCI), pre-fracture living situation (independent at home, at home with assistance for activities of daily living, institutionalized care facility), pre-fracture mobility (freely mobile without aids, mobile with one aid, mobile with two aids or frame, indoor mobility but outdoor immobile, no functional mobility), type of fracture (based on the OTA classification: OTA 31A for trochanteric fractures or OTA 31B for femoral neck fractures), and type of surgical procedure (sliding hip screw, proximal femoral nail anti-rotation, hemiarthroplasty, cannulated hip screw, hip arthroplasty) [18].

### 2.4. Outcome Measures

The primary outcome of this study was in-hospital mortality. Secondary outcomes were postoperative complications (surgical and non-surgical complications), admission to an intensive care unit (ICU), hospital length of stay, hospital readmission within 30 days after discharge, 30-day mortality, 90-day mortality, and 1-year mortality. Surgical complications included wound infection, postoperative hemorrhage, or secondary surgical intervention, such as wound rinsing and prosthesis revision. Non-operative complications included thrombo-embolic events (cerebrovascular accidents, deep venous thrombosis, and pulmonary embolisms), cardiac complications (myocardial infarction, arrhythmia, and congestive heart failure), pneumonia, urinary tract infection, delirium, pressure ulcer, need for blood transfusion, and urinary retention. Data on mortality were acquired by consulting the municipal citizen registry, and data on complications, when diagnosed by an attending physician, were extracted from electronic medical records.

### 2.5. Statistical Analysis

Statistical analysis was performed using SPSS for Windows 20.0 (IBM, Chicago, IL, USA). Differences between patients admitted before (pre-cohort) and after (post-cohort) the implementation of NOM within a palliative care context were analyzed using descriptive statistics. Continuous variables were tested for differences between groups with an unpaired *t*-test or Mann–Whitney U test, depending on normality. Normality was tested using the Shapiro–Wilk test. All categorical and dichotomous data were tested with a chi-square test. Descriptive statistics have been presented as mean with standard deviation or median with interquartile ranges (IQRs), depending on the distribution. For all statistical tests, *p*-values less than or equal to 0.05 are considered significant.

## 3. Results

### 3.1. Patient Demographics

In total, 1263 patients presented at the ED with a hip fracture. After exclusion, 1035 were included in the analysis. A total of 550 patients were included in the pre-cohort, and 485 patients were included in the post-cohort (Figure 1). The study population had a median age of 82 years (IQR 76–87), consisted of 688 females (66.5%), and had a median BMI of 24.0 (IQR 21.7–26.7). A femoral neck fracture was diagnosed in 588 (56.8%) of the patients, whereas 447 (43.2%) patients sustained a trochanteric fracture. The majority of 666 patients (64.3%) lived independently at home without additional care before admission, 206 patients (19.9%) lived at home with activities for daily living (ADL) support, and 163 patients (15.7%) were admitted to an institutionalized care facility (Table 1). Regarding the patients who were excluded from this analysis for receiving NOM within a palliative care context, these patients (*n* = 71) had a median age of 86 (IQR 82–91), 46 (64.8%) were female, 43 (60.6%) had a prior diagnosis of dementia, and the median CCI was 6 (IQR 5–7). Prior to the fracture, 46 (64.8%) patients lived in an institutionalized care facility, and only 4 (5.6%) patients could mobilize freely without aids (Table 2).

In the pre-cohort, 379 patients (68.9%) were female, and the post-cohort consisted of 309 female patients (63.7%) (*p* = 0.086). The post-cohort had significantly fewer patients diagnosed with dementia than the pre-cohort (72 (15%) vs. 116 (21%), *p* = 0.010). No significant difference between the pre-cohort and post-cohort was measured in the living situation and pre-fracture mobility before the hip fracture. In addition, there were no significant differences between the two cohorts at baseline regarding age, BMI, fracture type, or surgical procedure.

### 3.2. Mortality and Postoperative Complications

After the implementation of NOM within a palliative care context, no statistically significant difference was observed in in-hospital mortality (2.9% vs. 1.4%, *p* = 0.139). Additionally, the 30-day (6.4% vs. 4.7%, *p* = 0.281), 90-day (10.9% vs. 10.3%, *p* = 0.763), and 1-year (22.4% vs. 20.2%, *p* = 0.404) mortality follow-up periods also showed no statistical significance in mortality between the two cohorts (Table 3). Significantly more postoperative hemorrhages occurred in the post-cohort (0.2% vs. 1.9%, *p* = 0.018). Admissions to the ICU showed a significant decrease in the post-cohort (3.5% vs. 1.2%, *p* = 0.025). There were no significant differences in the incidence of other complications, readmissions, or hospital length of stay (Table 3). In total, 8 out of 71 (11.3%) patients died in the hospital who opted for NOM within a palliative care context. The 30-day, 90-day, and 1-year mortality were 57 (80.3%), 63 (88.7%), and 68 (95.8%), respectively. The median number of days until death for patients who opted for NOM within a palliative care context was 9 (IQR 5-22) days after hip fracture (Table 2).

## 4. Discussion

This study retrospectively analyzed elderly patients admitted to the trauma-geriatric ward after OM. The results of this study showed no significantly lower mortality rate or fewer postoperative complications for the post-cohort after the introduction of NOM within a palliative care context for hip fracture patients. However, significantly fewer operatively treated demented patients and significantly fewer ICU admissions in the post-cohort were observed. In addition, the NOM group showed the majority of patients opting for this management after hip fracture dying within 30 days (80.3%) with a median time till death of 9 (5–22) days.

This study observed an in-hospital mortality risk of 2.9% pre-implementation and 1.4% post-implementation, which corresponds with recent studies showing in-hospital mortality rates for geriatric hip fracture patients ranging from 1.5% to 5.0% [12,14,15]. Although the difference between the two cohorts in this study was not significant, a cautious trend to a lower in-hospital mortality can be seen in the percentage of in-hospital deceased patients. Subsequently, there are indications of a lower mortality rate with the absence of high-risk patients opting for NOM within a palliative care context. One-year mortality rates of 22.4% pre-implementation and 20.2% post-implementation also showed no significant decrease but did show lower mortality rates than recent literature, which ranges from 23.2% to 35.1% [19,20,21,22]. Compared with a 2018 cohort study in our center with a reported mortality rate of 27.0%, this study found substantially lower 1-year mortality rates [19]. This could be explained by the recent introduction of trauma-geriatric units and their subsequent improvement over the years [22,23,24]. One possible explanation for the minor impact on mortality rates could be that the clinical outcomes of the post-cohort were affected by the COVID-19 pandemic since these two periods largely coincided. The interference of the pandemic cannot be ruled out and possibly led to an overestimation of mortality, especially in the post-cohort since the geriatric population is particularly at risk of dying from COVID-19 [25,26,27]. Recent studies report that COVID-19 more than doubles the 90-day mortality rate following hip fracture and show a 30-day mortality rate of 34% in hip fracture patients with a COVID-19 infection [28,29]. Due to limitations in ascertaining the cause of death for patients who did not die within the hospital setting and privacy constraints that prevent access to such information from the municipal citizen registry, it was deemed unreliable to include these values in this study.

In the post-cohort (14.8%), there were significantly fewer patients with a prior diagnosis of dementia compared to the pre-cohort (21.1%). The incidence of the post-cohort is lower than earlier studies, showing an incidence of 20-28%, which is more in line with the incidence of the pre-cohort [19,24,30,31]. In recent literature, a high percentage (73%) of patients diagnosed with dementia opted for this NOM within a palliative care context after hip fracture, which probably explains the significant decrease in demented patients undergoing OM [10]. In a survey to investigate the general public’s view on life-sustaining treatment in the case of dementia, 72.9% expressed a preference for a peaceful passing and 68.9% expressed a preference for their partner to have a peaceful passing [32]. Therefore, it is possible that patients with dementia or those who care for them are more likely to opt for NOM within a palliative care context.

Although dementia has also been identified as a risk factor for early mortality after hip fracture, this does not imply that all dementia patients are at high risk of adverse outcomes after OM since this is a heterogeneous population with a wide range of physical and cognitive conditions resulting in a variable outcome [33]. Therefore, in the dementia population, it remains essential to include individual risk assessments in the decision-making process.

A possible explanation that the post-cohort did not show fewer postoperative complications could be due to the introduction of an automated complication registration method in our hospital in January 2021, as it is previously studied that automation of the registration process results in a rise in the incidence of registered complications without the increase in relative complications [34].

The postoperative incidence of secondary hemorrhage even increased significantly in the post-cohort. This result may be due to an increase in the usage of direct oral anticoagulants (DOACs) over the last few years [35]. Using a DOAC will not result in an unnecessary surgical delay in our center. There is evidence that wound complications, including secondary hemorrhage, have a higher incidence in geriatric hip fracture patients using DOACs [36].

Contrarily, ICU admissions showed a significant decrease in the post-cohort to 1.2%, which is also lower than that of previously reported incidence rates of 3–4%, which is again more in line with the incidence of the pre-cohort of 3.5% [30,37]. It seems possible that the incidence of patients with severe adverse outcomes requiring ICU admission after OM significantly decreased due to the identification of frail patients performed in acute settings resulting in a decrease in OM in the frailest patients. However, as previously stated, the post-cohort largely coincided with the COVID-19 pandemic. Therefore, it is important to consider that the decrease in ICU admissions during this period may be attributable to the limited availability of ICU beds; consequently, reducing the proportion of geriatric hip fractures admitted to the ICU [38].

One of the strengths of this study, since NOM within a palliative care context is relatively new in hip fracture management, is being the first to investigate the impact of the introduction of NOM within a palliative care context on operatively treated hip fracture patients in terms of mortality and morbidity. The main limitation was this study’s retrospective nature, resulting in difficulty in acquiring follow-up data for geriatric patient populations and, therefore, minor complications after admission or cause of death. Consequently, information on postoperative outcomes after discharge was only available if patients revisited the hospital. Furthermore, an additional limitation arises due to the inherent variability in treatment techniques for hip fractures, which depend on numerous fracture- and patient-related factors These variations may lead to differences in postoperative rehabilitation protocols and complication rates. Nevertheless, we opted to analyze the operatively treated group as a whole because our primary objective was to assess whether the overall cohort was influenced by the inclusion of very frail geriatric patients who likely opted for NOM within a palliative care context since its implementation. Another limitation could be the possible underpowering of this study due to the limited amount of available data post-implementation. However, this study has described the largest possible patient cohort following the implementation of NOM within a palliative care context in this center, representing the most comprehensive dataset within the specified timeframe. In addition, changes in management and protocols over time could also have affected the primary and secondary outcomes. Therefore, potential confounding due to the effect of time could exist. Lastly, this study only collected clinical data as outcome measures without functional or psychological outcomes. Future studies with prospective designs could give more insights into a possible improvement in these outcomes.

With the results of this study, it is tempting to speculate that in the post-cohort, there is no direct decrease in the overall frailty of the operatively treated hip fracture population. A decrease in total numbers was described; however not a statistically significant decrease in in-hospital mortality, there are indications that those with a high risk of short-term adverse outcomes are more likely to opt for NOM within a palliative care context. Furthermore, since patients opting for NOM within a palliative care context stay significantly shorter in the hospital and often (35.0%) return directly from the ED to their place of origin, fewer complex hip fracture patients are admitted to the trauma-geriatric ward, which further decreases the burden of care [11]. It is worth mentioning that although costs should not influence the choice of treatment, the introduction of NOM within a palliative care context has significantly lowered healthcare costs compared to OM, primarily due to shorter median hospital length of stay (3 (IQR 2–5) vs. 6 (IQR 4–9) days) and costs related to surgery and readmissions [39]. However, most importantly, with the introduction of NOM within a palliative care context, a shift in thinking from a disease-oriented to a patient-goal-oriented paradigm is ensured. This will provide better person-centered care for geriatric patients with limited life expectancy.

## 5. Conclusions

The introduction of NOM within a palliative care context for geriatric hip fracture patients did not result in a significantly lower in-hospital mortality rate, fewer postoperative complications, and hospital readmissions in the surgically treated geriatric hip fracture population. However, NOM within a palliative care context is considered a more patient-centered treatment modality by frail geriatric hip fracture patients with reduced pre-fracture mobility and very limited life expectancy.

## Figures and Tables

**Figure 1 jcm-13-02012-f001:**
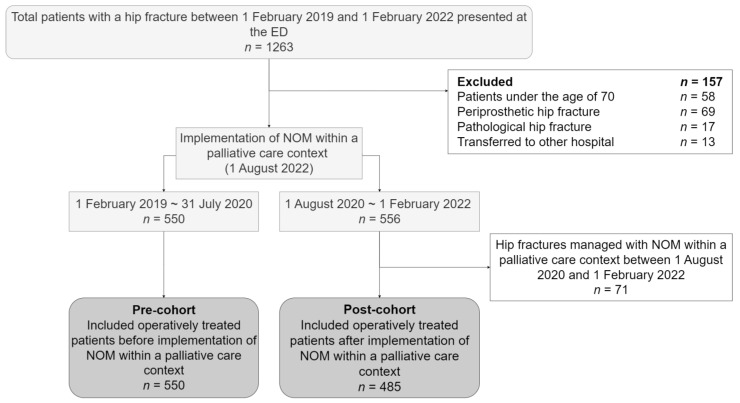
Flow chart of the selection process of included patients who underwent operative treatment for a hip fracture. ED = emergency department; NOM = non-operative management.

**Table 1 jcm-13-02012-t001:** Baseline characteristics of hip fracture patients who underwent operative treatment.

Baseline Variable	Data Missing	Total (*n* = 1035)	Pre-Cohort (*n* = 550)	Post-Cohort (*n* = 485)	*p*-Value
Age (in years)	0 (0)	82 (76–87)	82 (77–88)	82 (76–87)	0.893
Female sex	0 (0)	688 (66.5)	379 (68.9)	309 (63.7)	0.086
Prior diagnosis of dementia	0 (0)	188 (18.2)	116 (21.1)	72 (14.8)	0.010
BMI in kg/m^2^	27 (2.6)	24.0 (21.7–26.7)	23.9 (21.6–26.2)	24.2 (21.9–27.0)	0.095
CCI	0 (0)	5 (4–6)	5 (4–6)	5 (4–6)	0.373
Living situation before fracture	0 (0)				0.151
Home, independent		666 (64.3)	341 (62.0)	325 (67.0)	
Home, with ADL care		206 (20.0)	112 (20.4)	94 (19.4)	
Institutionalized care facility		163 (15.7)	97 (17.6)	66 (13.6)	
Pre-fracture mobility	2 (0.2)				0.147
Freely without aids		467 (45.1)	243 (44.2)	224 (46.2)	
Outdoors with 1 aid		47 (4.6)	27 (4.9)	20 (4.1)	
Outdoors with 2 aids or frame		496 (47.9)	272 (49.5)	224 (46.2)	
Indoor, but immobile outside		18 (1.7)	4 (0.7)	14 (2.9)	
No functional mobility		5 (0.5)	3 (0.5)	2 (0.4)	
Fracture type	0 (0)				0.314
Femoral neck fracture		588 (56.8)	304 (55.3)	284 (58.6)	
Trochanteric fracture		447 (43.2)	246 (44.7)	201 (41.4)	
Surgical procedure	0 (0)				0.136
Sliding hip screw		69 (6.7)	41 (7.5)	28 (5.8)	
Proximal femoral nail anti-rotation		443 (42.8)	246 (44.7)	197 (40.6)	
Hemiarthroplasty		462 (44.6)	237 (43.1)	225 (46.4)	
Cannulated hip screw		6 (0.6)	4 (0.7)	2 (0.4)	
Total hip arthroplasty		55 (5.3)	22 (4.0)	33 (6.8)	

All variables are in total amount (percentage) or median (interquartile range, IQR). BMI = body mass index; CCI = Charlson Comorbidity Index; ADL = activities of daily living.

**Table 2 jcm-13-02012-t002:** Characteristics and outcomes of hip fracture patients who underwent non-operative management within a palliative care context.

NOM Cohort	Data Missing	Total (*n* = 71)
Age (in years)		86 (82–91)
Female sex		46 (64.8)
Prior diagnosis of dementia		43 (60.6)
BMI in kg/m^2^	13 (18)	21.9 (19.7–24.2)
CCI		6 (5–7)
Living situation before fracture		
Home, independent		9 (12.7)
Home, with ADL care		16 (22.5)
Institutionalized care facility		46 (64.8)
Pre-fracture mobility	5 (7)	
Freely without aids		4 (5.6)
Outdoors with 1 aid		6 (8.5)
Outdoors with 2 aids or frame		29 (40.8)
Indoor, but immobile outside		19 (26.8)
No functional mobility		8 (11.2)
Admission in hospital		43 (60.6)
Hospital length of stay (in days)		3 (2–5)
Mortality		
In-hospital		8 (11.3)
30-day		57 (80.3)
90-day		63 (88.7)
1-year		68 (95.8)
Time till death (in days)		9 (5–22)

All variables are in total amount (percentage) or median (interquartile range, IQR). BMI = body mass index; CCI = Charlson Comorbidity Index; ADL = activities of daily living.

**Table 3 jcm-13-02012-t003:** Patient outcomes of operatively treated hip fracture patients.

Patient Outcomes	Pre-Cohort (*n* = 550)	Post-Cohort (*n* = 485)	*p*-Value
Mortality			
In-hospital	16 (2.9)	7 (1.4)	0.139
30-day	35 (6.4)	23 (4.7)	0.281
90-day	60 (10.9)	50 (10.3)	0.763
1-year	123 (22.4)	98 (20.2)	0.402
Patients with complications	286 (52)	272 (56)	0.190
Surgical complications	28	31	
Wound infection	24 (4.4)	20 (4.1)	0.878
Secondary hemorrhage	1 (0.2)	9 (1.9)	0.018
Re-intervention	3 (0.5)	2 (0.4)	1.000
Non-surgical complications	418	416	
Thrombo-embolic	8 (1.5)	8 (1.6)	0.807
Cardiac	42 (7.6)	40 (8.2)	0.731
Pneumonia	48 (8.7)	41 (8.5)	0.912
UTI	37 (6.7)	35 (7.2)	0.807
Delirium	149 (27.1)	138 (28.5)	0.627
Pressure ulcer	26 (4.7)	30 (6.2)	0.336
Anemia	70 (12.7)	77 (15.9)	0.154
Urinary retention	30 (5.5)	40 (8.2)	0.083
Sepsis	8 (1.5)	7 (1.4)	1.000
Admission to ICU	19 (3.5)	8 (1.2)	0.025
Readmission	26 (4.7)	32 (6.6)	0.223
Hospital length of stay (in days)	6 (4-9)	6 (5-9)	0.053

All variables are in total amount (percentage) or median (interquartile range, IQR). UTI = urinary tract infection; ICU = intensive care unit.

## Data Availability

The datasets used and/or analyzed during the current study are available from the corresponding author upon reasonable request.

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
