# Peer review of "The Impact of Implementation of Palliative, Non-Operative Management on Mortality of Operatively Treated Geriatric Hip Fracture Patients: A Retrospective Cohort Study"

_jcm, 2024, doi:10.3390/jcm13072012_

Round 1

Reviewer 1 Report

Comments and Suggestions for Authors

abstract:  i suggest to mention how many patients involved,  mean follow-up, and mortality rate

Line 61 : 218, hip fractures "is it for coding system?"

Table 1 , 2 ,3 : please provide Asa Score whenever possible / history of cardiac disease if possible

Table 2 : there was only 8 patients who had no functional mobility among 71 patients (63 patient still have some degree of functional mobility) and there was no ASA evaluation (and lowest BMI was 19.7)  >> how criteria was chosen for non operative ? 

authors compared pre-cohort and post cohort for operative ,but why authors did not compare global surgical groupe versus NOM ? 

what was the rational to seperate pre-cohort and post cohort? the aim of the study should be focused on surgical treatment (1035 operative vs 71 NOM) 

it's better to make table and compare between these group

it's not clear for me why authors focused on pre-cohort and post cohort as all 2 groups were surgically treated ?  isn't better to compare only 485 patients(post cohort) with the NOM 71 patients ? 

please note that mortality rate was 22 ,20 % in both group of surgically managed patients , while non operative had 95%  (Line 189-191)  >> why no statistical analysis performed ?

why trochanteric fracture and femoral neck fracture was not seperated? and why femoral neck fracture (Garden) was not mentioned , as triple screw has lower surgical risk than hemiarthroplasty ?  please remember that trochanteric fracture is usually more painful than femoral neck 

so i suggest to to subgroup analysis in others tables  (adding ASA score)

1- global surgical treatment (pre-cohort + post cohort) based on femoral neck fracture  versus trochanteric fracture  and compared it with NOM   > measuring complications (same as table 3)

2- Give more details of table 2 (NOM) type of fracture and give subgroup analysis with operative treatment on same groupe (femoral neck or trochanteric fracture) 

remember to add ASA score / complications  for NOM (same as table 3 for operative treatment) 

3- for table 2 : compare pre-fracture with post fracture ( as i imagine most patients were dependent after fracutre) and who lived in home became institutionalized in care facility, this will be interesting to mention

Please don't forget to add type of fracture as i'm not convinced that intertrochanteric fracture should not be operated ,  spinal anesthesia with gamma nail in 15 minutes can facilitate ambulation (only 8 /71 in this group was not ambulated!)

Give more details for NOM protocole , what happens to these patients ? how were they followup , pain mangement , were there bedridden ? or can be transferred to armchair for sitting ? as we use rarely NOM for femoral neck fracture in high risk patients and we transfer them for sitting with pain management 2-5 days after fracture to avoid bedridden complications

please also add complication of NOM patients (decubitus) > as mortaility was 95% at 1 year 

please provide more details how NOM decision was shared with patients and family 

Give limitation of the study in a paragraph

conclusion is misleading "the introduction of NOM within a palliative care context for geriatric hip fracture 321 patients did not result in a significantly lower in-hospital mortality rate, fewer postoperative complications, and hospital readmissions in the surgically treated geriatric hip fracture population"  authors should mention high mortality rate 

in summery: subgroup analysis should be performed based on type of fracture and compare it with NOM    (globally and based on fracture type)

Pre NOM vs Post nom should be compared

complication of NOM 

Author Response

Comment 1: abstract:  I suggest to mention how many patients involved,  mean follow-up, and mortality rate

Response 1: Thank you for your feedback. We agree with your comment and have accordingly revised the abstract to include the number of patients (page 1, line 18). The in-hospital and 1-year mortality rates were already provided in the abstract. Due to the retrospective nature of the study involving file searching in electronic patient records, patients were not followed up beyond hospital admission, and thus only data from the initial hospital stay were utilized. Mortality data was obtained by consulting the municipal citizen registry.

Comment 2: Line 61 : 218, hip fractures "is it for coding system?"

Response 2: Thank you for pointing out this lack of clarity. We have duly addressed the issue regarding code 218, and it has been rectified in the manuscript for enhanced clarity (page 2, lines 66-73).

Comment 3: Table 1 , 2 ,3 : please provide Asa Score whenever possible / history of cardiac disease if possible

Response 3: Thank you for your suggestion. We agree with the benefit of the ASA-score to give insights in the health status of the population and meeting the Loggers et al. frailty criteria for Hip fracture patients.1 However, we have decided to use the Charlson Comorbidity Index instead of the ASA score, since it offers a more comprehensive insight into the health status of the population. Also, the inter-observer inconsistency of the ASA classification was taken into account. Due to the subjectiveness of the ASA classification and a significant level of disagreement exists, even between qualified specialists.2,3 The Charlson Comorbidity Index is a scoring system used to assess the severity of comorbid conditions in patients. The index assigns a weighted score to various comorbidities based on their association with mortality. These comorbidities include 17 variables such as heart disease, diabetes, cancer, and liver disease, among others.

1Loggers SAI, Willems HC, Van Balen R, et. al. FRAIL-HIP Study Group. Evaluation of Quality of Life After Nonoperative or Operative Management of Proximal Femoral Fractures in Frail Institutionalized Patients: The FRAIL-HIP Study. JAMA Surg. 2022 May 1;157(5):424-434. doi: 10.1001/jamasurg.2022.0089. PMID: 35234817; PMCID: PMC8892372.

2Mak PH, Campbell RC, Irwin MG; American Society of Anesthesiologists. The ASA Physical Status Classification: inter-observer consistency. American Society of Anesthesiologists. Anaesth Intensive Care. 2002 Oct;30(5):633-40. doi: 10.1177/0310057X0203000516. PMID: 12413266.

3Riley R, Holman C, Fletcher D. Inter-rater reliability of the ASA physical status classification in a sample of anaesthetists in Western Australia. Anaesth Intensive Care. 2014 Sep;42(5):614-8. doi: 10.1177/0310057X1404200511. PMID: 25233175.

Comment 4: Table 2 : there was only 8 patients who had no functional mobility among 71 patients (63 patient still have some degree of functional mobility) and there was no ASA evaluation (and lowest BMI was 19.7)  >> how criteria was chosen for non operative ?

Response 4: We agree that these characteristics may be remarkable in the context of historical non-operative treatments, which were typically reserved for either relatively young and healthy individuals with non-displaced femoral neck fractures or those with significant comorbidities rendering surgery non-beneficial, often leading to surgical refusal by anesthesia. However, the novel treatment approach of NOM within a palliative care context for very frail geriatric patients with limited life expectancy likely demonstrates distinct characteristics.

NOM within a palliative care context is offered through shared decision-making, guided by either the Frailty Criteria described by Loggers et al. ((Body Mass Index (BMI) of 18.5 kg/m2 or lower, Functional Ambulation Category (FAC) of 2 or lower pre-trauma, ASA score of 4 or 5)) or the clinical judgment of the attending surgeon. This clinical judgment is informed by the intuition of highly experienced trauma surgeons who frequently encounter such patients. It is worth noting that these patients always retain the option to undergo surgery if that is their preference. We believe that offering patients the choice of NOM within a palliative care context provides a more patient-centered approach.

Comments 5-7: what was the rational to seperate pre-cohort and post cohort? the aim of the study should be focused on surgical treatment (1035 operative vs 71 NOM)

it's better to make table and compare between these group

it's not clear for me why authors focused on pre-cohort and post cohort as all 2 groups were surgically treated ?  isn't better to compare only 485 patients(post cohort) with the NOM 71 patients ?

Response 5-7: The rationale for comparing the pre- and post-cohorts originates from our hypothesis that since NOM is now offered in a palliative care context to very frail geriatric hip fracture patients with very limited life expectancy, these patients are more inclined to opt for NOM, thereby possibly altering the composition of the operatively treated population. Consequently, the operatively treated population may no longer include the most frail patients, potentially yielding improved outcomes compared to before, when outcomes may have been skewed by the relatively poor outcome of these very frail geriatric hip fracture patients.

Comment 8: please note that mortality rate was 22 ,20 % in both group of surgically managed patients , while non operative had 95%  (Line 189-191)  >> why no statistical analysis performed ?

Prima geantwoord

Response 8: This study primarily focuses on the operatively treated population before and after implementation of NOM within a palliative care context. Hence, the statistical analysis was confined to a comparison between these two groups. Patients opting for NOM within a palliative care context are no longer included in the operatively treated group, potentially leading to different outcomes. Therefore, statistical analysis was performed to demonstrate whether significant differences existed between the operatively treated geriatric hip fracture population before and after the implementation of NOM within a palliative care context. Descriptive reporting of NOM data was included to provide supplementary information and enhance transparency.

Comments 9-15: why trochanteric fracture and femoral neck fracture was not seperated? and why femoral neck fracture (Garden) was not mentioned , as triple screw has lower surgical risk than hemiarthroplasty ?  please remember that trochanteric fracture is usually more painful than femoral neck so i suggest to to subgroup analysis in others tables  (adding ASA score)

1- global surgical treatment (pre-cohort + post cohort) based on femoral neck fracture  versus trochanteric fracture  and compared it with NOM   > measuring complications (same as table 3)

2- Give more details of table 2 (NOM) type of fracture and give subgroup analysis with operative treatment on same groupe (femoral neck or trochanteric fracture)

remember to add ASA score / complications  for NOM (same as table 3 for operative treatment)

3- for table 2 : compare pre-fracture with post fracture ( as i imagine most patients were dependent after fracutre) and who lived in home became institutionalized in care facility, this will be interesting to mention

Please don't forget to add type of fracture as i'm not convinced that intertrochanteric fracture should not be operated ,  spinal anesthesia with gamma nail in 15 minutes can facilitate ambulation (only 8 /71 in this group was not ambulated!)

Response 9-15:

Thank you for providing such valuable feedback. We acknowledge and agree with all the points you have raised. However, in our study, we focus on a novel form of NOM, with a palliative perspective. We believe that for patients with a very limited life expectancy, a more patient-centered approach is necessary. Operating on these patients and administering anesthesia results in postoperative pain and potential rapid mortality. Conversely, opting not to operate on a patient also results in pain, but foregoing surgery yields a comparable quality of life as the surgically treated population, as demonstrated by Loggers et al. Resulting in a potentially more peaceful last phase of life for these very frail geriatric patients with very limited life expectancy. Additionally, with effective pain management and emerging techniques such as the PENG-block, a peripheral nerve block, satisfactory analgesia outcomes can be achieved.

The focus of our study is on the operatively treated hip fracture population. Further elaboration on the outcomes of the NOM group is deemed irrelevant to our research question but certainly valuable in further research of our study group. NOM within a palliative care context represents a relatively new treatment option with a palliative perspective in our institution. Loggers et al. have demonstrated that patients opting for NOM treatment experience a non-inferior quality of life compared to those choosing surgery in the final stages of life. We have provided a more detailed explanation of NOM within a palliative care context in the methods of the manuscript to ensure clarity and understanding among readers regarding its palliative, non-operative nature (pages 2-3, lines 77-102). Given its palliative nature, patient outcomes such as complications and ambulation are no longer monitored, and these patients are likely to pass away soon.

The aim was to investigate the entire cohort of hip fracture patients rather than focus on specific fracture types. We do not expect that excluding very frail geriatric hip fracture patients would yield any significant new insights into the differences between femoral and trochanteric fractures. Our research question centers on whether the option of NOM within a palliative care context for very frail geriatric hip fracture patients has influenced the results of the operatively treated group. We acknowledge that the fact this is a new treatment modality may not have been adequately clarified in the manuscript and have provided a more comprehensive and explicit explanation (pages 1-2, lines 38-45).

Comments 16-18: Give more details for NOM protocole , what happens to these patients ? how were they followup , pain mangement , were there bedridden ? or can be transferred to armchair for sitting ? as we use rarely NOM for femoral neck fracture in high risk patients and we transfer them for sitting with pain management 2-5 days after fracture to avoid bedridden complications

please also add complication of NOM patients (decubitus) > as mortaility was 95% at 1 year

please provide more details how NOM decision was shared with patients and family

Response 16-18: Thank you for these suggestions and we agree with these comments. Therefore, we have provided a more extensive description of the NOM within a palliative care context protocol in the manuscript (pages 2-3, lines 77-102). The option is presented through an extensive Shared Decision Making (SDM) conversation, frequently involving family members, wherein both the outcomes of surgery and NOM within a palliative care context are thoroughly discussed. Approximately 25% of patients presenting to the ED engage in an extensive shared decision-making conversation, during which both treatment options are thoroughly discussed. Ultimately, approximately half of these patients choose NOM within a palliative care context, while the remaining half express a preference for surgical intervention. Hence, patients always retain the option to undergo surgery if they wish to. Satisfactory analgesia results are acquired with new effective modalities such as the PENG-block for NOM patients. Given the palliative approach, patients often return directly back to their place of origin and no outcomes such as ambulation and bedridden complications are monitored or registered. However, the experience of these patients was assessed through proxy-reported interviews, which concluded that proxies, in general, were satisfied with the chosen option of palliative, non-operative management.1

1Nijdam TMP, Laane DWPM, Spierings JF, et alProxy-reported experiences of palliative, non-operative management of geriatric patients after a hip fracture: a qualitative studyBMJ Open 2022;12:e063007. doi: 10.1136/bmjopen-2022-063007

Comment 19: Give limitation of the study in a paragraph

Response 19: Limitations have been addressed in a “strengths and limitations” paragraph in the discussion section.

Comment 20: conclusion is misleading "the introduction of NOM within a palliative care context for geriatric hip fracture 321 patients did not result in a significantly lower in-hospital mortality rate, fewer postoperative complications, and hospital readmissions in the surgically treated geriatric hip fracture population"  authors should mention high mortality rate

Response 20: Thank you for pointing this out. However, we believe the phrasing in the conclusion reflects our research question, which centers on whether the introduction of NOM within a palliative care context affects the mortality and postoperative complications of the operatively treated group. The high mortality of NOM patients lies beyond the scope of our research question and is therefore not included in the conclusion. In addition, in this context, NOM is not curative in nature but rather in a palliative setting, where the outcome is not solely focused on survival but on the quality of life and comfort in the final stage of life.

Reviewer 2 Report

Comments and Suggestions for Authors

I congratulate the authors on their well-written manuscript which shows the extensive work that was put into the analysis of hip fracture patients at their institution. There are a few questions I would like to ask:

Line 68: Non-operative management (NOM) within a palliative care context was introduced on 1/8/2020. Before this timepoint, were all patients treated with surgery regardless of frailty? If not, what was the demography of patients that did not receive surgery before introduction of NOM at your institution?

Line 108: Please clarify the terms "pre-cohort" and "post-cohort" in your Statistical analysis section. For instance: ...patients admitted before ("pre-cohort") and after ("post-cohort") the implementation of NOM...

Figure  1: Box below "Implementation of NOM within a palliative care context" on the right: Timeframes are wrong (01-08-2022 - 01-02-2022 should be 01-08-2020 - 01-02-2022)

Table 1: There is an inevitable high heterogeneity in treatment techniques to address hip fractures depending on many fracture-related and patient-related factors. Different surgeries may come with different post-operative rehabilitation protocols and occurrences of complications and should either be analyzed in sub-categories or listed as limitation of your study. Although percentages of surgery types seem to be comparable in both timeframes, there is still an unexpected low number of patients treated with cannulated hip screws as potential minimal-invasive option for frail patients and undisplaced fractures. Is there a reason this surgery type is not regularly used in your institution? Could you give more information on fracture types, i.e. Garden classification, AO classification?

Line 222: How was NOM decided? Did frail patients (or relatives) actively opt for NOM after informed consent and both OM and NOM presented to them? In this case, were there frail patients that opted out of NOM and received surgery?

Line 229: The decrease you are mentioning is only in total numbers and not statistically proven, i.e. there is no statistically significant decrease. With a p=0.139 between the pre- and post-cohort, it may be more correct to describe it as a cautious trend to a lower in-hospital mortality.

Line 241: You discuss the interference of the COVID-pandemic on potential parameters of the post-cohort. Could there have been an effect of the COVID-pandemic on the availability of ICU beds, which ultimately led to a lesser percentage of hip fracture patients receiving those beds?

Line 271: This may be a good explanation for a higher secondary hemorrhage incidence. Do you have data on how many patients received DOAKs in both timeframes?

Line 274: What is an unnecessary delay after using DOAKs in your center? Do you have data on waiting times between admission and surgery in general and in patients receiving DOAKs? Do you perform e.g. arthroplasty surgery in a patient with active DOAK use?  Do you routinely measure, for instance, Anti-Xa in those patients with Rivaroxaban before surgery? Are there any cut-off levels?

Author Response

Comment 1: Line 68: Non-operative management (NOM) within a palliative care context was introduced on 1/8/2020. Before this timepoint, were all patients treated with surgery regardless of frailty? If not, what was the demography of patients that did not receive surgery before introduction of NOM at your institution?

Response 1: Before the implementation of NOM within a palliative care context, certain patients underwent non-operative treatment. These individuals typically comprised relatively young and healthy subjects with non-displaced femoral neck fractures, or those with significant comorbidities rendering surgery non-beneficial, often resulting in surgical refusal by the anesthesia team. Thus, this category included a distinct form of NOM, different from the novel treatment approach of NOM within a palliative care context. Nonetheless, we have included the demographics of these patients in the manuscript to offer supplementary insights and improve transparency. It will be clearly elucidated that these two groups are not directly comparable, but their inclusion in our manuscript provides additional contextual information.

Comment 2: Line 108: Please clarify the terms "pre-cohort" and "post-cohort" in your Statistical analysis section. For instance: ...patients admitted before ("pre-cohort") and after ("post-cohort") the implementation of NOM...

Response 2: Thank you for pointing this out. The terms "pre-cohort" and "post-cohort" have been adjusted in the Statistical Analysis section of the manuscript (page 3, line 135).

Comment 3: Figure  1: Box below "Implementation of NOM within a palliative care context" on the right: Timeframes are wrong (01-08-2022 - 01-02-2022 should be 01-08-2020 - 01-02-2022)

Response 3: Thank you for pointing this out. The timeframes have been updated in the manuscript's figure (page 5, line 172).

Comment 4: Table 1: There is an inevitable high heterogeneity in treatment techniques to address hip fractures depending on many fracture-related and patient-related factors. Different surgeries may come with different post-operative rehabilitation protocols and occurrences of complications and should either be analyzed in sub-categories or listed as limitation of your study. Although percentages of surgery types seem to be comparable in both timeframes, there is still an unexpected low number of patients treated with cannulated hip screws as potential minimal-invasive option for frail patients and undisplaced fractures. Is there a reason this surgery type is not regularly used in your institution? Could you give more information on fracture types, i.e. Garden classification, AO classification?

Response 4: Thank you for providing this valuable feedback, and we appreciate your suggestion to address it as a limitation of our study. Accordingly, we have updated the strengths and limitations paragraph in the discussion section of the manuscript (page 10, lines 336-343).

We opted to consider the operatively treated group as a whole because our primary objective was to assess whether the overall cohort would demonstrate improved outcomes after excluding the NOM within a palliative care context population. However, we ensured comparability between pre- and post-groups by including fracture and surgery type as baseline variables. Our aim was to examine the entire cohort of hip fracture patients rather than focusing solely on specific fracture types and corresponding surgery types. We anticipate that excluding very frail geriatric hip fracture patients would not yield significant new insights into the differences between femoral and trochanteric fractures and their respective surgery types.

Furthermore, our institution rarely utilizes cannulated hip screws due to the risk of femur head necrosis. While they are less invasive, the potential need for subsequent intervention, such as hemiarthroplasty or total hip arthroplasty, led us to minimize their use. Additionally, we utilize the OTA 31-A classification for trochanteric fractures and OTA 31-B classification for femoral neck fractures, rather than the Garden or AO classification systems. We acknowledge this lack of clarity and have included this categorization in our manuscript to ensure consistency in reporting (page 3, lines 114-115).

Lastly, retrospectively assessing all radiographs would entail a considerable amount of work without necessarily providing substantial additional insights.

Comment 5: Line 222: How was NOM decided? Did frail patients (or relatives) actively opt for NOM after informed consent and both OM and NOM presented to them? In this case, were there frail patients that opted out of NOM and received surgery?

Response 5: Thank you for your question and addressing this lack of clarity. Therefore, we have incorporated a clarification into the methods of our manuscript (pages 2-3, lines 77-102). NOM within a palliative care context is offered to the most frail geriatric hip fracture patients, guided either by the Frailty Criteria described by Loggers et al. or the clinical judgment of the attending surgeon. This decision is informed by the intuition of highly experienced trauma surgeons who frequently encounter such patients. The option is presented through an extensive Shared Decision Making (SDM) conversation, frequently involving family members, wherein both the outcomes of surgery and NOM within a palliative care context are thoroughly discussed. Approximately 25% of patients presenting to the Emergency Department engage in an extensive shared decision-making conversation, during which both treatment options are presented. Ultimately, approximately half of these patients choose NOM within a palliative care context, while the remaining half express a preference for surgical intervention. Hence, patients always retain the option to undergo surgery if they wish to.

Comment 6: Line 229: The decrease you are mentioning is only in total numbers and not statistically proven, i.e. there is no statistically significant decrease. With a p=0.139 between the pre- and post-cohort, it may be more correct to describe it as a cautious trend to a lower in-hospital mortality.

Response 6: Thank you for this valuable suggestion. We agree that it is important to formulate it cautiously and have attempted to do so. The use of a ‘trend’ may be avoided and we favored the sentence you used in the comment i.e. ‘a decrease in total numbers however not statistically significant decrease’ (page 9, lines 269-270 and page 10, lines 356-357).

Comment 7: Line 241: You discuss the interference of the COVID-pandemic on potential parameters of the post-cohort. Could there have been an effect of the COVID-pandemic on the availability of ICU beds, which ultimately led to a lesser percentage of hip fracture patients receiving those beds?

Response 7: We extend our gratitude for the valuable suggestion provided. We agree with the comment and have consequently excluded the mention of the decrease in ICU admission rate from our conclusion (page 11, lines 376-378). Moreover, we have included a paragraph in the discussion section, explaining on the potential rationale behind the observed decrease in ICU admissions (page 10, lines 323-327).

Comment 8: Line 271: This may be a good explanation for a higher secondary hemorrhage incidence. Do you have data on how many patients received DOAKs in both timeframes?

Response 8: Thank you for acknowledging our statement. While we agree that this could provide additional substantiation to our statement, we currently do not have easily accessible data.

Comment 9: Line 274: What is an unnecessary delay after using DOAKs in your center? Do you have data on waiting times between admission and surgery in general and in patients receiving DOAKs? Do you perform e.g. arthroplasty surgery in a patient with active DOAK use?  Do you routinely measure, for instance, Anti-Xa in those patients with Rivaroxaban before surgery? Are there any cut-off levels?

Response 9: Thank you for your inquiries and interest in this matter. Historically, surgery for hip fractures was delayed for at least 24 hours according to protocol. However, in our center, it was determined by expert consensus not to delay surgery for patients using DOACs. These patients undergo surgery as soon as possible. However, this is beyond the scope of the current manuscript and thereby data was not registered at the emergency department and therefore not applicable for this study.

Round 2

Reviewer 1 Report

Comments and Suggestions for Authors

authors answered all inquiries very clearly and i thank them for their efforts, 

i have no issues to add , and i think it's ok for publication